# Proteotoxicity: A Fatal Consequence of Environmental Pollutants-Induced Impairments in Protein Clearance Machinery

**DOI:** 10.3390/jpm11020069

**Published:** 2021-01-25

**Authors:** Shweta Devi, Jong-Joo Kim, Anand Prakash Singh, Surendra Kumar, Ashish Kant Dubey, Sandeep Kumar Singh, Ravi Shankar Singh, Vijay Kumar

**Affiliations:** 1Systems Toxicology and Health Risk Assessment Group, CSIR-Indian Institute of Toxicology Research, Lucknow 226001, India; sweta.kamal123@gmail.com; 2Department of Biotechnology, Yeungnam University, Gyeongsan, Gyeongbuk 38541, Korea; kimjj@ynu.ac.kr; 3Division of Cardiovascular Disease, The University of Alabama at Birmingham (UAB), 1720 2nd Ave South, Birmingham, AL 35294-1913, USA; apsingh@uabmc.edu; 4Cytogenetics Lab, Department of Anatomy, All India Institute of Medical Sciences, New Delhi 110029, India; surendrakhedarcbt@gmail.com; 5Department of Neurology, SGPGIMS, Lucknow 226014, India; ashish.icom@gmail.com; 6Department of Medical Genetics, SGPGIMS, Lucknow 226014, India; sandeepcbt@gmail.com; 7Department of Biochemistry, Microbiology & Immunology, University of Saskatchewan, Room 4D40, Health Sciences Building, 107 Wiggins Road, Saskatoon, SK S7N 5E5, Canada

**Keywords:** protein misfolding, molecular chaperones, co-chaperone, JUNQ, IPOD, INQ, protein degradation, environmental pollutants

## Abstract

A tightly regulated protein quality control (PQC) system maintains a healthy balance between correctly folded and misfolded protein species. This PQC system work with the help of a complex network comprised of molecular chaperones and proteostasis. Any intruder, especially environmental pollutants, disrupt the PQC network and lead to PQCs disruption, thus generating damaged and infectious protein. These misfolded/unfolded proteins are linked to several diseases such as Parkinson’s disease, Alzheimer’s disease, Huntington’s disease, and cataracts. Numerous studies on proteins misfolding and disruption of PQCs by environmental pollutants highlight the necessity of detailed knowledge. This review represents the PQCs network and environmental pollutants’ impact on the PQC network, especially through the protein clearance system.

## 1. Introduction

Eukaryotic cells have a well-organized, tightly regulated protein quality control (PQC) system. This quality control system includes the molecular chaperones, ubiquitin/proteasome-dependent protein degradation, and autophagy machinery (target and uptake of non-native conformer in the spatial compartments) that consistently monitors and maintains the conformational state of cellular proteins. In eukaryotic cells, this surveillance is done by various structurally unrelated molecular chaperones [1,2,3,4,5,6]. The members of these families have been termed heat shock proteins (HSPs). HSPs are classified according to their molecular weights (Hsp40, Hsp60, Hsp70, Hsp90, Hsp100, and small HSPs). Under stress conditions, these chaperones form a complex co-operative network and try to maintain the conformational state of cellular proteins. However, the proteins (misfolded/unfolded) that are unidentified by the PQC network exhibit toxicity for the cellular environment. To cope up with aberrant proteins, cells have several different ways to restore the integrity of a healthy proteome such as degradation via autophagy or ubiquitin-dependent proteolysis [7,8,9]. Error in the breakdown of aberrant polypeptides may result in protein aggregation which leads to pathological conditions such as Alzheimer’s disease (AD), Parkinson’s disease (PD), Huntington’s disease (HD), cataracts, and diabetes [10,11,12,13,14,15,16,17,18].

Several intracellular and extracellular stress factors can affect the integrity of the native protein, which causes severe damage either by loss of structure or the gain of toxic activity [19,20]. Recent studies suggest the involvement of environmental toxicants such as pesticides, air pollutants, and heavy metals. These environmental pollutants disrupt the PQCs network either by affecting molecular chaperones or proteasomal degradation machinery [21,22]. Sometimes these aggregates are prompt to sequestration in distinct subcellular compartments. These compartments are juxtanuclear quality control compartment (JUNQ), insoluble protein deposit (IPOD), and intranuclear quality control compartment (INQ). Recent studies indicate that co-chaperones are involved in the transfer process of aberrant proteins towards these compartments [1,23,24]. This review summarizes the essential PQC networks and the environmental pollutants which adversely affect the PQC system especially protein clearance machinery.

## 2. Essential Network of PQC

### 2.1. Aberrant Protein Processing by Molecular Chaperones

Many intracellular and extracellular factors like mutations, change in pH, temperature, and reactive oxygen species (ROS) levels, cause misfolding in proteins [25,26,27]. These misfolded proteins are recognized by molecular chaperones. These chaperones interact with misfolded conformers either by hydrophobic or electrostatic interactions followed by ATP hydrolysis. ATP hydrolysis causes conformational changes in the chaperone’s apical domain that forces the client protein to enter the chaperones’ cavity. Chaperones give enough time for a protein to fold by slowing down the rate of ATP hydrolysis [28]. The eukaryotic chaperone in tailless complex polypeptide 1 ring complex (TRiC) is an obligate chaperone for many proteins. In TRiC, folding occurs outside of the chamber [29,30,31,32].

The total time of insertion of misfolded proteins into the chaperone cavity depends upon the extent of protein’s hydrophobicity as well as ATP binding affinity of the molecular chaperone’s smaller subunit. For instance, TRiC has eight subunits and each of them has a different charge and different hydrophilic amino acids, therefore it shows a different affinity for client proteins and ATP. Hence, TRiC shows selective client binding and different ATP binding affinity [28,31,32,33,34,35,36,37].

The selectivity of client protein also depends on the type of co-chaperone. For instance, Hsp70 works with a co-chaperone known as J-protein (also called Hsp40). These J-proteins are known to have the J-domain. Human cells have 41 different kinds of J-proteins and all these have different selectivity for different client proteins. These J-domains, generally, are embedded inside the protein core which is exposed to stress conditions (denatured or misfolded protein) [34,35,36,37,38,39]. Therefore, exposure to these motifs indicates the presence of stress factors that affect the labile proteins. These stress damaged proteins interact with Hsp70 either by hydrophobic or electrostatic interactions. This binding affinity also depends upon the motion (opening and closing) of Hsp70’s α-helical lid located on the C-terminal of β sheet sub-domain [28,34,35,36,37,38,39]. This lid opening and closing is mediated by ATP hydrolysis [36,40]. The lid sub-domain C-terminal tail also shows an additional binding affinity for the client protein that increases the chances of rebinding when necessary [34,35,36,37,41,42].

Studies also show that client proteins after interaction with Hsp70 remain less structured for the entire duration of their folding. This strategy prevents proteins from undergoing a globular hydrophobic collapse [36,37,43]. Both stresses damaged as well as native proteins follow this trend during their synthesis [36,38]. Biophysical studies reveal that several different conformations are adopted by the lid subdomain and by the tail to provide accommodation for heavier and folded segments of the proteins. Different conformations of subdomain lid and tail allow Hsp70 to bind with unfolded polypeptides, folding intermediates, and near-native conformations by using a variety of different binding modes [34,35,36,37,39,44,45].

### 2.2. Molecular Co-Chaperone Interlinks Chaperones and Degradation Pathway

A potential misfolded protein is recognized and trapped by chaperones. These chaperones alleviate the toxicity of the misfolded conformers either by sequestering them or by modulating their conformation [11,19,46,47]. Chaperones are potentially engaged with the folding of newly synthesized proteins, their translocation, and the refolding of unfolded client proteins. In addition to these roles, chaperones also target misfolded conformers for degradation and transfer them in spatial compartments with the help of co-chaperones [1,11,19,47,48]. These co-chaperones have a combination of the domains that function in the ubiquitin-proteasome system (UPS) and chaperone interacting domain.

Co-chaperone carboxyl terminus of Hsc70-interacting protein (CHIP) and Bcl-2-associated athanogene-1 (BAG1) initially identified proteins having this structural feature and this finding gives an idea about direct communication between folding and degradation. BAG1 has a BAG domain and a ubiquitin-like (UBL) domain. BAG domain has Hsp70 nucleotide-binding activity and the UBL domain binds with 26S proteasome. CHIP has a tetratricopeptide repeat (TPR) domain, this TPR domain binds with Hsp70 and Hsp90 and a U-box with E3 ligase activity [49,50,51,52,53,54,55,56,57]. These findings suggest that the co-chaperone facilitates the delivery of substrates to protein degradation machinery. CHIP, Parkin, HSJ1a, BAG6, and UFD2B co-chaperones all have a domain for both interactions with a chaperone and function in the UPS (Table 1) [46].

### 2.3. PQC Compartments: Final Steps of Protein Processing

Cellular inclusion is an integral part of the PQC system which seems to be conserved from yeast to mammalian cells [58,59,60]. These compartments are well known as the juxtanuclear quality control compartment (JUNQ), insoluble protein deposit (IPOD), and intranuclear quality control compartment (INQ). In these systems, misfolded proteins get sequestered for further processing either for refolding or degradation. Sorting of these misfolded proteins depends on a variety of characteristics such as misfold status, ubiquitination status, or chaperone interaction.

Chaperone interaction with misfolded proteins like Hsp110/Sse1, Hsp70/Ssa1, labels them for sequestration in Q-bodies and further targets them to the JUNQ compartment [1]. The IPOD sequestered proteins are generally non-ubiquitinated, immobile, non-diffusing, and insoluble. Under severe stress conditions, these IPODs act as a reservoir for misfolded protein [2] while JUNQ sequestered misfolded proteins are ubiquitinated and remain in a soluble state [2,61]. Studies have demonstrated that the JUNQ compartment forms near the surface of the nucleus during the time of proteasome impairment (Figure 1). Proteins are in a soluble state in the JUNQ compartment so they can diffuse easily. Thus, during stress conditions, JUNQ acts as a reservoir for misfolded protein clearance or refolding [2].

If ubiquitination is blocked, then misfolded proteins are transported inside the nucleus where they get sequestered in INQ (Figure 1) with the blockage of CDC48 [62]. These misfolded proteins are also targeted for ubiquitination by San1 and then degraded by the proteasome inside the nucleus (Figure 1) [1]. Recent studies suggest that chaperones also target misfolded proteins toward different compartments, like Sis1 and Hij1, which are necessary for the Q-body formation and their maturation.

A family of chaperones called the hook family has members: Btn2 and Cur1. Both members have roles in the formation of IPOD and JUNQ [1,61]. Btn2 and Cur1 target misfolded proteins into these compartments in a small HSP (Hsp42)-dependent manner [61]. Hsp42 is a small ATP independent HSP, involved in the Q-body formation, during severe stress conditions. Hsp42 along with Hsp26 gets hyper-activated and co-localized with IPOD and other PQC sites. Hsp70 family member Ssa1/2 has a role in the refolding of misfolded proteins in Q-bodies and coalescence of Sis1 and Q-body targets the misfolded proteins toward INQ [2,63,64,65].

Hsp90 family member Hsc82 refolds proteins in Q-body and helps during the coalescence of Q-body with Ssa1/2 [63]. Co-chaperone HOP acts as a sorting factor and targets the aberrant proteins towards JUNQ. Chaperones Sis1 and Hsp40 also act similarly. Sis1 with V-SNARE binding protein Btn2 targets misfolded proteins to JUNQ and Sis 1 with Cur1 (curing of Ure3) targets misfolded proteins to IPOD. Chaperones Sis1 and Hsp70 together target the misfolded proteins towards INQ [24,38,61,66]. A small HSP like Hsp26 delivers the proteins towards spatial compartments under heat shock. Hsp42 with Btn2 targets proteins toward IPOD [67]. Sorting is also dependent upon ubiquitination.

From several studies [24,68], it has been shown that ubiquitin-labeled proteins are directed towards JUNQ. This compartment is highly concentrated with 26S proteasome subunits (lacking in IPOD). This suggests that the misfolded protein degradation in JUNQ happens in a ubiquitin-dependent manner [68]. Furthermore, these JUNQ compartments are also rich in disaggregating chaperones, therefore, facilitating misfolded protein disaggregation and refolding [2].

The location of IPOD (near the vacuoles, autophagic vesicles) suggests that IPOD sequestered misfolded proteins are targeted for degradation by auto-phagosome. Even though there is no functional link present between IPOD and autophagosome [2], evidence suggests the presence of an autophagic marker Atg8 in IPOD. This suggests that IPOD sequesters misfolded proteins that are cleared by the autophagic pathway [69,70,71].

## 3. Endoplasmic Reticulum-Associated PQC

The endoplasmic reticulum (ER) is the site for protein folding. The ER has multiple types of folding machinery that ensure that only the correctly folded proteins reach the site of action. When an unfolded protein reaches the ER lumen, Hsp70 co-chaperone Hsp40 binds to an unfolded protein [72,73]. In case of an error in the folding of proteins due to oxidation, mutation, or environmental stresses, there is a blockage of their release from the lumen of the ER into the cytosol. These misfolded proteins are required to relocate from the ER to the cytosol for degradation by the proteasome. These ER-associated misfolded proteins are processed by three pathways, namely: ER-associated degradation: ERAD-L (for misfolded ER lumenal proteins), ERAD-M (for ER-membrane proteins with a misfolded membrane domain), and ERAD-C (for ER membrane proteins with a misfolded cytosolic domain). All these pathways occur in the cytosol using specific types of machinery [74,75]. Ubiquitination of misfolded proteins occurs in the cytosol and they are further targeted towards proteasomal degradation. This process of PQC followed by proteasomal elimination of the misfolded protein is termed ER-associated degradation (ERAD), and it depends on an intricate interplay between the ER and the cytosol [76].

The mammalian system uses a combination of different proteins to accomplish the ER-linked PQC and ERAD. Proteins, namely BiP (GPR-78), ERdj6, ERdj1, ERdj2, ERdj3, ERdj4, ERdj5, OST (oligosaccharyl transferase complex), Glucosidase-I, Glucosidase-II, Calnexin, Calreticulin (CRP55), UGGT1, ERMan1, EDEM1, EDEM2, EDEM3, OS-9, XTP3-B, POMT1, POMT2, PDI, Hsp70, and Hsc70 are the modifiers, lectins, and chaperones helping in the ER-associated PQC while ERAD machinery comprises several different proteins, each having a specific function.

E3 ubiquitin ligase MARCH6 (TEB-4) helps during the membrane passage and ubiquitination. HRD, gp78, Sel-1L, Herp, Derlin-1, Derlin-2, and Derlin-3 help in the substrate re-translocation and ubiquitination. UBE1, UBE1L2 (MOP-4), Ubc6, HsUBC6e, Ube2G1, Ube2G2 (UBC7), Derlin-1, Derlin-2, and Derlin-3 are involved with the substrate ubiquitination process. Cdc48 (cell division cycle 48) complex’s protein including p97 (VCP), UFD1, NPL4, and Ubxd8 (FAF2) are involved with the extraction of the substrate from the ER and further target it for the ubiquitination process. PLC1, PLC2, (Ubiquilin3, A1Up) hHR23a, and hHR23b are the shuttling factors that shuttle the ubiquitinated substrate towards the proteasome. Substrates of the ERAD pathways are marked with ubiquitin chain with the help of ubiquitin-conjugating enzymes (E1, E2, and E3). After ubiquitination, the tagged protein is identified by 26S proteasome (RPN1, RPN10, and RPN13), deubiquitinated RPN11 (PSMD14), and further degraded, this pathway is important for the degradation of ERAD aberrant protein [76,77,78,79].

## 4. Environmental Pollutants: Effect on PQC

### 4.1. Protein Degradation Machinery

Proteolysis mediated by ubiquitin plays an important role in several basic cellular processes. Disruption in this pathway leads to the accumulation of misfolded proteins in the cell. As proteolysis is accompanied by the ubiquitination process, the target protein is first processed by the Ubiquitin-Activating Enzyme (E1). This enzyme activates the ubiquitin by hydrolyzing the ATP. The resultant of this step is an E1-thiol-ester∼ubiquitin intermediate. The activated E1-ubiquitin moiety is further transferred to E2, a ubiquitin-conjugating enzyme [80,81]. E2 either conjugates the ubiquitin to the target protein or transfers the E1-ubiquitin moiety to an E3-ubiquitin intermediate. The E2 active site is comprised of the presence of the ubiquitin-conjugating enzymes (UBC) domain (required for the binding with E3) and a ubiquitin-binding Cys residue [82]. Currently, many E2 have been identified, among them Ubc4 and Ubc5 are involved with the targeting of misfolded/short-lived proteins. Another E2, Ubc3/Cdc34, is involved with the targeting of phosphorylated substrates. The number, variety, and function of different E2s in mammalian species are much greater and are still being explored. The E3, ubiquitin-protein ligase enzyme is engaged with the recognition of substrates of the ubiquitin system. E3 ligases are binds with both E2 and substrates. In most cases, E3 act as a scaffold to bring together the E2 and substrate; this process allows the transfer of ubiquitin moiety from E2 to the substrate [77,79,83,84,85,86].

After the ubiquitination process, the protolysis work of the target protein is conducted by 26S proteasome machinery. The 26S machinery is comprised of a 20S core and 19S cap. Their functional state attains a cylinder-like structure, where the 19S cap recognizes the substrate while the degradation work is achieved by 20S core moiety. The core of 20S has been shown to have a proteolytic active site. This complex is made up of four to seven numbered rings, in which two outer rings are known as α ring while the central side ring is known as β-ring. The proteolytic activity is carried only by β-ring [87,88,89,90,91,92]. In archaea, these β-rings have shown the chymotrypsin-like, trypsin-like, and peptidyl-glutamyl-peptide hydrolyzing activity [93].

### 4.2. Effect of Pesticide on Proteasome Degradation Machinery

The wide use of pesticides causes huge human encounters daily. These pesticides may enter the human body in different ways. Though the human body can neutralize these chemical effects, epidemiological studies demonstrate their toxicity linked to several diseases especially proteinopathies such as PD [94,95,96]. However, the mechanism behind their toxicity is still elusive.

Rotenone, an organic pesticide, has been extensively studied for PD pathogenesis. Rotenone treatment in rats shows a dopaminergic neuronal loss, ROS generation, and motor dysfunction [97,98]. Its toxicity on mitochondrial complex-1 elevates oxidative stress [99,100]. Though rotenone causes oxidative stress, its role in causing cellular death is far more than assumed. UPS is the site for protein degradation. Several studies highlight rotenone toxicity in the UPS [21,101,102,103,104,105]. One study has shown that rotenone causes mitochondrial inhibition, oxidative stress, microtubule assembly, and increased degradation of proteasomal subunits (mechanisms of rotenone-induced proteasome inhibition).

Benomyl, dieldrin, diethyldithiocarbamate, endosulfan, and ziram significantly inhibit the proteasome activity in human neuroblastoma cell line SK-N-MC^U^ [104]. Another study conducted in China reveals that the nine most frequently used pesticides (paraquat, rotenone, chlorpyrifos, pendimethalin, endosulfan, fenpyroximate, tebufenpyrad, trichlorphon, and carbaryl) induce morphological changes in the mitochondria at low concentrations. These pesticides are known to induce mitochondrial fragmentation. Interestingly, they also significantly suppress the activity of 26S and 20S proteasome [106]. Another study conducted on wild type mice depicts the ability of low dose paraquat, maneb, and chlorpyrifos to affect the 26S proteasome complex. Paraquat and its co-treatment with maneb or chlorpyrifos were found to be toxic and inhibited all soluble proteasomal expression of 26S proteasomal subunits [107]. Ziram, a fungicide, induces the primary ventral mesencephalic cell death by interfering with the UPS (26S) activity. Mechanistically, ziram inhibits the ubiquitin-activating enzyme (E1), thus altering UPS function [108].

### 4.3. Effect of Heavy Metal on Proteasome Degradation Machinery

Heavy metals are naturally occurring element present in the Earth’s crust, but haphazard human activity alters the geochemical balance. Huge human exposure causes a harmful effect. Earlier studies show that their exposure can cause various disorders including metabolic and neurological disorders also associated with lethal diseases [109,110]. These heavy metals also affect the PQC system especially the proteasome machinery. Cadmium is a divalent cation that significantly impairs proteasome activity. This impairment leads to an increase in the formation of soluble and insoluble ubiquitinated proteins [22]. Another study reveals that gallium containing compound inhibits cellular 26S proteasome and chymotrypsin-like activity of the purified 20S proteasome with IC_50_ values of 17 and 16 μmol/L [111].

Organic copper complexes also potentially inhibit the chymotrypsin-like activity of purified 20S proteasome. In human leukemia cells, proteasome inhibition occurs within 15 min after organic copper complexes exposure [112]. Cobalt induces apoptosis in the lung by disturbing the ubiquitin-proteasome pathway. In U-937 cells and human alveolar macrophages (AMs), exposure to cobalt chloride (CoCl_2_) induces apoptosis and accumulation of ubiquitinated proteins. Exposure to CoCl_2_ also suppresses the activity of the proteasome, although the inhibitory effect of CoCl_2_ (1000 μM) on proteasome activity is not as potent as that of MG-132 (proteasome inhibitor) (100 μM) [113].

Arsenic (As) is a metalloid that is rarely found as a free element in the natural environment. Arsenic has many pharmacological properties; however, due to its toxic property it has not gained much attention. Arsenic also causes perturbation of proteasome degradation machinery. Exposure of NaAsO_2_ on HEK293T, GM00637, IMR90, HEK293T, and NB4 APL cells shows significant impairment of enzymatic activities of ZnF containing E3 ubiquitin ligases. As_2_O_3_ exposure on NB4 APL cells also shows similar effects. Arsenate exposure on NIH3T3 and DTC25 cells result in the compromised functions of p97 and proteasome [114]. Arsenic exposure to HT22 cells leads to inactivation of proteasome machinery, damage in the electron transport chain, and an increase in oxidative stress [115].

### 4.4. Effect of Air Pollution on Proteasome Degradation Machinery

Epidemiological studies consistently document the relationship between increased risk of human disease with ambient air pollutants and particulate matter (PM) [116]. Several mechanistic studies provide insight for their deleterious outcome towards humans but all this information is still not enough to elaborate their mode of toxicity at the cellular and molecular level. Yet, few studies exist that exhibit their toxicity on the PQC system.

Exposure to two distinct experimental aerosols, diesel exhaust and secondary organic aerosol (DE and SOA), produces a rapid (within 30 min post-exposure) and consistent decrease in proteasome activity in two different peripheral blood cell types [117]. Though their immediate exposure causes a decrease in proteasome activity, their mechanism is still lacking. Both DE and SOA are directly associated with oxidative and nitrosative stress hence, the two might be directly associated with the decreased proteasomal activity. They are likely to play a role in response to PM in the air [118].

Cigarette smoke extract (CSE) treatment on human alveolar epithelial cells induced time and dose-dependent cell death. It also elevated the intracellular ROS and increased the level of carbonylated and polyubiquitinated proteins. Further, high doses of CSE impaired all proteasomal activities, while low CSE concentrations significantly inhibited only the trypsin-like activity of the proteasome in alveolar and bronchial epithelial cells [119].

### 4.5. Effect of Environmental Pollutants on ER-Associated PQC

Several studies reveal the toxicity of environmental pollutants on ER-associated PQC. These environmental pollutants, including pesticides and heavy metals, and certain nanoparticles show significant involvement in the disturbance of ER-associated PQC. Pyrethroids like insecticides are widely used on insects and pests in agriculture and public health. In recent years, pyrethroid-associated toxicity has risen dramatically. Pyrethroids exposure is also associated with the increasing incidences of neurodegenerative diseases [120,121,122,123,124,125].

Recent studies reveal their link with apoptosis, but the mechanism is still lacking. Some studies have found that deltamethrin pesticide initiates the apoptotic cascade through interaction with Na^+^ channels, leading to calcium overload and initiation of the apoptotic cascade. Studies have identified the role of calpain and the ER stress pathway as mediators of deltamethrin-induced apoptosis [126]. Exposure of permethrin on 3T3-L1 adipocytes and 3T3-L1 cells induces ER stress via changing the intracellular calcium level also elevates the level of ER stress marker protein [127].

Another class of pesticides, the organophosphates are associated with hampering the quality control of the ER. Treatment of chlorpyrifos (CPF) of 50 μM or 100 μM on human choriocarcinoma trophoblastic cells JEG-3 cells for 24 h induces an increase of ER stress-related proteins, such as GRP78/BiP and IRE1α and also causes the mRNA splicing of phospho-eIF2α and XBP1 [128]. Like pesticides, heavy metals are also potentially toxic for ER exhibiting toxicity via inducing unfolded protein response in ER (UPR). Treatment of cadmium (Cd) in *Saccharomyces cerevisiae* triggers the UPR via activating Ire1. Silver nanoparticles (AgNP), widely used in the medical field, are also reported to cause cytotoxicity via induction of apoptosis. Treatment of AgNP on human MCF-7 and T-47D breast cancer cells shows the accumulation of misfolded/aggregated proteins and induction of ER stress via activation of UPR [129].

During the ERAD process, the ERAD adaptor protein Ubx2 plays a role as the bridging factor. Ubx2 transports the misfolded proteins from the ER to the cytosol for ubiquitylation and proteasomal degradation. Recently, a study represented the Cd toxicity in *Saccharomyces cerevisiae*. Cd exposure in *Saccharomyces cerevisiae* induces the ER stress and causes the upregulation of ER stress-responsive genes (*HAC1*, *IRE1*, *ERO1*, and *PDI1*), heat shock responsive genes (*HSP104* and *HSP60*), ERAD pathway genes (*DOA10*, *CDC48*, *HRD1*, and *YOS9*), and proteasome regulators (*UBI4* and *RPN4*) [130].

### 4.6. Effect of Environmental Pollutant on Autophagy

PQC is accomplished by molecular chaperones and targeted protein degradation [131]. Targeted protein degradation is primarily performed by the ubiquitin-proteasome system (UPS). Recently, it has been established that the autophagic-lysosomal pathway (ALP) plays an important part in the removal of the aggregated form of misfolded proteins [132]. Hence, the UPS and the ALP are the final part of the defense against proteotoxicity.

Heavy metals and metalloids are proven to have a hazardous impact on humans [109]. Arsenic (As) is generally present in the environment in its two oxidative forms such as arsenous acid (H_3_AsO_3_) and its salts, and arsenic acid (H_3_AsO_5_) and its salts. Arsenic trioxide (As_2_O_3_) is majorly used as a pharmaceutical agent for its antitumor properties. Exposure of As_2_O_3_ leads to the activation of numerous intracellular signal transduction pathways such as ROS generation, mitochondrial disruption, and caspase activation resulting in the induction of apoptosis [133].

Current research has revealed that under As stress condition, autophagy could be initiated in place of apoptosis, as one study showed that As_2_O_3_ significantly induces autophagic cell death in leukemic cell lines [134,135]. Chromium in combination with cadmium can potentially induce autophagy. Co-treatment of cadmium and chromium in hematopoietic stem/progenitor cells (HSPC) induces autophagic morphologies. Toxic compounds that can potentially induce autophagy are summarized in Table 2.

## 5. Open Question and Future Prospect

To answer the question of cellular protein misfolding/aggregation, it is required to understand the basic knowledge of the PQC system and its inhibitors. The constant increase in evidence of environmental pollutants and protein misfolding diseases highlights the need to understand the link between environmental pollutants and the pathogenesis of several proteins misfolding diseases. Accordingly, we can study the significantly potent toxic effect of these environmental pollutants on human health. The fragile nature of the misfolded proteins and their escape from the PQC system suggest that a cell is either incapable to distinguish between toxic or non-toxic misfolded protein or, is unable to combat the proteotoxicity associated with toxic misfolded protein species.

Limited knowledge of PQCs significantly affects the development of novel therapeutics. There is also a need to understand the level of protein misfolding patterns and accordingly respond to PQCs in specific patients for a better understanding of the PQC system and the development of therapeutics. It is also important to decipher whether the formation of spatial compartments (for restoring the misfolded protein) are serving as protective ensembles or are acting as toxic entities. If they are working as toxic entities, then what initial events trigger their formation and how a cell can prevent their formation remains unattended.

## Figures and Tables

**Figure 1 jpm-11-00069-f001:**
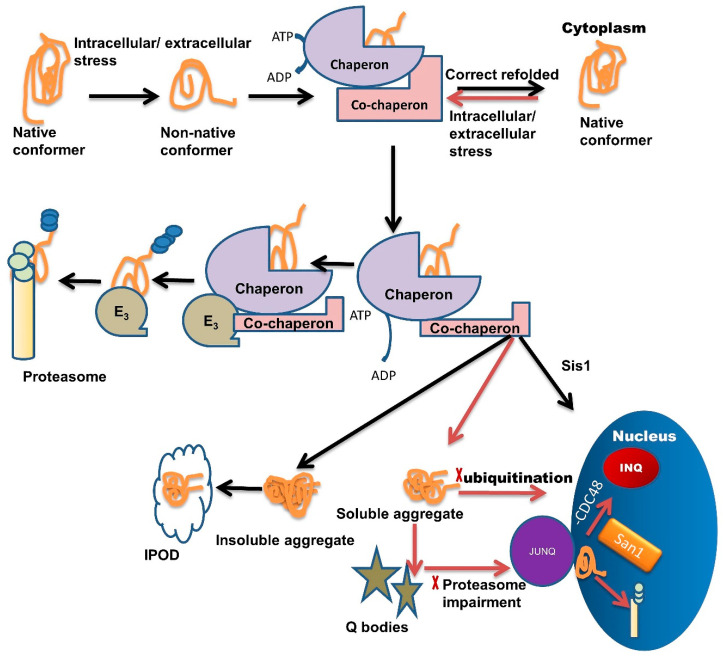
Multiple roles of molecular chaperones contributing to the management of misfolded proteins and cellular fitness. Exposure to intracellular or extracellular stress causes the formation of non-native protein conformer, which is identified and trapped by molecular chaperones through adenosine triphosphate (ATP) hydrolysis. Molecular chaperones refold misfolded proteins to their native conformation. In the case of protein folding failure, chaperones transfer misfolded proteins from chaperones to the UPS with the help of co-chaperones. E3 ubiquitin-ligase domains in co-chaperones recruit E3 ligases. E3 ligases further direct the ubiquitinated tagged protein towards the proteasome machinery for degradation and tagging them for ubiquitination. In some cases, these misfolded species are not recovered or are degraded by the PQC system due to the formation of soluble or insoluble aggregates. Insoluble aggregates sequester inside the IPOD while soluble aggregates/stress-damaged proteins sequester inside the Q-bodies. Under very high-stress conditions, soluble aggregates tend to JUNQ sequestration while some misfolded protein tends to sequester in INQ via Hsp70 co-chaperone Sis1 present in the nucleus. Inside the nucleus these misfolded aggregates either sequester in INQ by the blocking of CDC48 or target for degradation through ubiquitination via nuclear ubiquitin ligase San1.

**Table 1 jpm-11-00069-t001:** N-terminal and C-terminal domains of co-chaperones [46].

Co-Chaperone	Amino Terminal	Function	Carboxy Terminal	Function
CHIP	TPR repeats	Binds with Hsp70/Hsp90	U-box	E3 activity
UFD2B	Chaperone binding	Binds with Hsp70 co-chaperones DnaJc7	U-box	E3 activity
Parkin	UBL domain	Proteasome binding	RING domain	E3 activity
BAG6	UBL domain	Proteasome binding	BAG domain	Binds Hsp70
HSJ1a	J-domain	Binds Hsp70	UIM1, UIM2	Ub-binding

BAG6 (Bcl-2-associated athanogene-6), BAG (Bcl-2-associated athanogene), CHIP (Carboxyl terminus of Hsc70-interacting protein), HSJ1a (Homo sapiens J-domain protein), TPR (Tetratricopeptide repeat), UBL (Ubiquitin-like).

**Table 2 jpm-11-00069-t002:** List of toxic compounds that induce autophagy.

Compound	Experimental Model	Autophagic Effects	Ref.
As_2_O_3_	Human leukemia cells	Autophagic cell death, by activation of the MEK/ERK pathway.	[134,136,137]
	HL60 leukemia cells	Autophagic cell survival, in the earlier period of treatment; apoptosis and/or autophagic cell death, in the later period of treatment.	
	Human leukemia cells	Autophagy as clearance mechanism of the fusion oncoprotein PML/RARA.	
Cadmium	Skin epidermal cells	Autophagy induction by the increase in LC3-II formation and the GFP-LC3 puncta cells.	[138]
Chlorpyrifos	Human neuroblastoma SH-SY5Y cells	Induction of apoptosis.	[139]
Lindane	Mouse Sertoli cells	Increase in autophagy by disrupting mitochondria, autolysosomes, and ROS.	[140,141]
Nickel	Human bronchial epithelial BEAS-2B cells	Increase in autophagy by disrupting mitochondria and ROS.	[142]
Paraquat	Human neuroblastoma SH-SY5Y cells	Increase in autophagy by disrupting and inducing mitochondria, ER-stress, ROS.	[143,144]
Rotenone	Human neuroblastoma SH-SY5Y cells	Increase and impairment increase in mitochondria/lysosome destabilization, ROS.	[145]
Fipronil	Human neuroblastoma SH-SY5Y cells	Increase in autophagy by disrupting and inducing mitochondria, ROS.	[139]
Glyphosate	Neuronal differentiated PC12 cells	Increase in autophagy by disrupting mitochondria.	[146]

Reactive oxygen species (ROS), Endoplasmic reticulum (ER) stress, Mitogen-activated protein kinase/ERK kinase (MEK), extracellular-signal-regulated kinase (ERK).

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
