# Peer review of "Proteotoxicity: A Fatal Consequence of Environmental Pollutants-Induced Impairments in Protein Clearance Machinery"

_jpm, 2021, doi:10.3390/jpm11020069_

Round 1
Reviewer 1 Report
In the review “Environmental pollutants-induced disruption to protein clearance machinery generates proteotoxicity”, the Authors focused on the eukaryotic protein quality control (PQC) system comprised of molecular chaperones and intracellular proteins degradation via ubiquitin/proteasome-dependent pathway or autophagy. The impact of environmental pollutants (such as pesticides, heavy metals, and air pollutants) on the PQC system is discussed.
Although the subject is interesting, especially due to a toxic effect of the environmental pollutants on human health, some important concerns need to be addressed before the manuscript is ready for publication.
Major comments:
- In the text, reference numbers should be placed before, not after, the punctuation (for example [1], or [1].) according to the Instructions for Authors placed in the Journal Menu. The Authors placed all reference numbers after the punctuation, making some sentences/fragments difficult to understand.
- Sections 4.1-4.3: The Authors discuss the effect of environmental pollutants on proteasome degradation machinery. Major components of the ubiquitin-proteasome system should be very briefly described in the text, including E1, E2, and E3 enzymes (ubiquitin-activating enzyme, ubiquitin-conjugating enzyme, and ubiquitin ligase, respectively) involved in the ubiquitination process as well as 26S proteasome complexes. Peptidase activities of the 20S proteolytic core of the 26S proteasome (chymotrypsin-, trypsin- and caspase-like activities) should be mentioned.
- Terms “chaperone” and “chaperones” should be given in the same way throughout the text and figures. The Authors use both “chaperone” and “chaperon” as well as “”chaperones” and “chaperons”. I suggest to use “chaperone/chaperones”.
- Both terms “ubiquitination” and “ubiquitinylation” are used in the manuscript. I suggest to use “ubiquitination” throughout the text as well as “ubiquitinated” instead of “ubiquitinylated”.
- Disease names should be given in the same way throughout the text and in the Abbreviations section (Parkinson's disease, Alzheimer's disease, Huntington’s disease or Parkinson disease, Alzheimer disease, Huntington disease).
- Page 2, lines 38-39: It should be “This quality control system includes the molecular chaperones, ubiquitin/proteasome-dependent protein degradation, and autophagy machinery” instead of “This quality control system includes the molecular chaperones, protein degradation, and autophagy machinery“.
- Page 2, lines 49-50: It should be “ubiquitin-dependent proteolysis” instead of “ubiquitin-dependent proteostasis”.
- Page 2, lines 68-69: The statement ”exposure to pH, temperature” is not clear enough.
- Page 3, line 76: There is no information about TRiC in Ref. [10]. This reference is for the bacterial chaperonin GroEL. I suggest to include another reference for the eukaryotic chaperonin TRiC (for example the review: Balchin D, Hayer-Hartl M, Hartl FU. In vivo aspects of protein folding and quality control. Science. 2016 Jul 1;353(6294):aac4354. doi: 10.1126/science.aac4354. PMID: 27365453).
- Page 3, lines 83-89: The fragment about Hsp70 and J-proteins should be checked and corrected (see also comments no. 11 and 12). I suggest to include a more recent reference on Hsp70 (for example: Mayer, M.P.; Gierasch, L.M. Recent advances in the structural and mechanistic aspects of Hsp70 molecular chaperones. J. Biol. Chem. 2019, 294, 2085–2097).
- Page 3, line 84: I suggest “a co-chaperone known as J-protein (also called Hsp40)” instead of “a co-chaperone known as J-protein”.
- Page 3, lines 88-89: The statement “Hsp70 co-chaperone and Hsp40 have a domain known as the J-protein binding domain ” is not clear. Hsp40 co-chaperones are collectively called J-proteins because they all contain a J-domain that is required for interaction with Hsp70. I suggest to check Ref. [13] as well as Frontiers in Neuroscience 2018, vol.11, article 743 for information about Hsp70 and Hsp40 interaction. Ref. [15] seems not adequate.
- Page 4, lines 113 and 115: Ref. [12] seems not adequate.
- Page 4, lines 124-126: Not all co-chaperones mentioned are included in Table 1.
- Page 5, lines 149-151: The statement “E3 ligases further degrade the misfolded proteins by tagging them for ubiquitinylation and direct them towards proteasome machinery” is not entirely correct because E3 enzymes (ubiquitin ligases) do not degrade proteins. Proteins are degraded by 26S proteasomes.
- Page 7, lines 240-242: The sentence beginning with “p97 (VCP)” is difficult to understand.
- Page 7, lines 244-246: I suggest to check and rewrite the sentence beginning with “26S proteasome (26S proteasome is mentioned as the proteasomal enzyme together with subunits of 19S regulatory complex and cytosolic PNGase).
- Page 8, lines 260-262: The statement “One hypothesis targets its downstream toxicity on UPS. UPS is the site for protein degradation.[55-57] Several studies highlight its toxicity in proteasomal dysfunction.” is not clear enough. One can conclude from this fragment that “Several studies highlight UPS toxicity in proteasomal dysfunction.
- Page 8, line 273: The statement “affect 26S proteasome subunit” is not clear enough. The 26S proteasome is a multisubunit complex.
- Page 8, line 277: The statement “E1 ligase” seems not correct although it is taken from Ref. [62]. I would recommend to use “E1 enzyme” or “ubiquitin-activating enzyme (E1)” (see also comment no.2).
- Page 8, lines 288-290: The statement “Another metal gallium inhibits cellular 26S proteasome and chymotrypsin-like activity of the purified 20S proteasome with IC50 values of 46, 27, and 16 μmol/L.[65]” is not clear enough. The IC50 values mentioned are those reported for three of five gallium complexes tested by [65].
- Page 11, lines 377-380: It is difficult to understand the statements “Current research has revealed that autophagy can be an alternative or accompanying process to apoptosis in As-exposed cells.[86] examined the capability of As2O3 in inducing autophagic cell death in leukemic cell lines: Their data demonstrate that As is a potent inducer of autophagy.[87]”. This fragment should be corrected (see also comment no. 1).
- Page 11, Table 2 (third column/Chlorpyrifos row): The statement “Increase in Autophagy by affecting autophagy” is not clear.
- Correcting and editing the English used in the manuscript (including the Graphical Abstract) is recommended (some suggestions are given in comments).
Minor comments:
- Page 2, line 52: It should be “cataract, and diabetes” instead of “Cataract, and Diabetes”.
- Page 2, line 67: It should be “Aberrant protein processing by molecular chaperones” instead of “Aberrant protein processing by molecular chaperon”.
- Page 3, line 101: It should be “native” instead of “naive”.
- Page 4, lines 117 and 119: It should be “BAG1” instead of “Bag1” (this name should be given in the same way throughout the text).
- Page 4, line 122: It should be “Hsp70 and Hsp90” instead of “HSP70 and HSP90”.
- Page 4, line 124: It should be “HSJ1a” instead of “HSJ1A”, “BAG6” instead of “BAG 6”, and “UFD2B” instead of “UFD2D”.
- Page 4, TABLE 1: First column (SN) is not necessary. It should be “Hsp70” instead of “HSP70”, “Hsp90” instead “HSP90”, “RING” domain” instead of “Ring domain”, “BAG domain” instead of “Bag domain”, and “HSJ1a” instead of “HSJ1A”.
- Page 5: Figure 1 should be placed within the section 2.3.
- Page 6, line 170: It should be “The IPOD sequestered proteins are” instead of “The IPOD sequester proteins are”.
- Page 7, lines 227-228: The term “ERAD” should be included in the text when it arrives for the first time (see line 220).
- Page 7, line243: I suggest “ubiquitinated substrate” instead of “ubiquitinylated substrate” (see comment no. 4).
- Page 7, line 246: It should be “associated” instead of “associate”.
- Page 8, line 256: I suggest a new paragraph from the sentence beginning with “Rotenone, an organic pesticide”.
- Page 10, line 334: It should be ‘pyrethroid-associated toxicity” instead of “pyrethroid associate toxicity”.
- Page 10, line 349: It should be “unfolded protein response in ER (UPR)” instead of “unfolded protein response in ER” (The Authors use the UPR abbreviation in the next sentence).
- Page 10, line 354: It should be “activation of UPR [80].” instead of “activation of unfolded protein response(UPR).[80]”.
- Page 10, line 355: I suggest a new paragraph from the sentence beginning with “During ERAD process”.
- Page 10, line 362: It should be “UBI4” instead of “UBI14” (There is an error in the abstract of Ref. [81] taken by the Authors as a source of information).
- Page 10, line 364: It should be “on autophagy” instead of “on Autophagy”.
- Page 11, line 379: It should be “As2O3” instead of “As2O3”.
- Page 11, Table 2 (first column): It should be “As2O3” instead of “AS2O3”.
- Page 11, Table 2 (third column): I suggest “the fusion oncoprotein PML/RARA” instead of “the fusion protein PML/RARA”.
- Page 11, Table 2 (third column/Chlorpyrifos row): It should be “Increase in autophagy” instead of “Increase in Autophagy”.
- Page 11, Table 2 (second column/Nickel row): I suggest “Human bronchial epithelial BEAS-2B cells” instead of “BEAS-2B”.
- Page 11, Table 2 (second column): I suggest “human neuroblastoma SH-SY5Y cells” instead of “SH-SY5Y”.
- Page 11, Table 2 (second column/Glyphosate row): I suggest “neuronal differentiated PC12 cells” instead of “PC12”.
- Page 12, line 407: I suggest “ABBREVIATIONS” instead of “ABBREVIATION”.
- Page 12, line 408: It should be “HSP, heat shock protein” or “HSPs, heat shock proteins” instead of “HSP, heat shock proteins”.
- Page 12, lines 408-415: I suggest an alphabetical order of abbreviations. “UPR” and “CSE” should be included.
- Page 12, line 414: It should be “DE, diesel exhaust” instead of “DE, Diesel exhaust”.

Author Response
First, we would like to express our sincere gratitude to the reviewers for their critical and insightful suggestions and comments. We revised the manuscript according to these suggestions. All amendments are written in blue in the revised manuscript.
Responses to Comments:
Reviewer 1:
In the review “Environmental pollutants-induced disruption to protein clearance machinery generates proteotoxicity”, the Authors focused on the eukaryotic protein quality control (PQC) system comprised of molecular chaperones and intracellular proteins degradation via ubiquitin/proteasome-dependent pathway or autophagy. The impact of environmental pollutants (such as pesticides, heavy metals, and air pollutants) on the PQC system is discussed.
Although the subject is interesting, especially due to a toxic effect of the environmental pollutants on human health, some important concerns need to be addressed before the manuscript is ready for publication.
Major comments:
- In the text, reference numbers should be placed before, not after, the punctuation (for example [1], or [1].) according to the Instructions for Authors placed in the Journal Menu. The Authors placed all reference numbers after the punctuation, making some sentences/fragments difficult to understand.
Ans. We thank the reviewer for the overall very positive evaluation. We have gone through the manuscript again and revised the manuscript. Punctuation are modified as per your suggestion.
- Sections 4.1-4.3: The Authors discuss the effect of environmental pollutants on proteasome degradation machinery. Major components of the ubiquitin-proteasome system should be very briefly described in the text, including E1, E2, and E3 enzymes (ubiquitin-activating enzyme, ubiquitin-conjugating enzyme, and ubiquitin ligase, respectively) involved in the ubiquitination process as well as 26S proteasome complexes. Peptidase activities of the 20S proteolytic core of the 26S proteasome (chymotrypsin-, trypsin- and caspase-like activities) should be mentioned.
Ans. Required suggestions were added in the revised manuscript.
- Terms “chaperone” and “chaperones” should be given in the same way throughout the text and figures. The Authors use both “chaperone” and “chaperon” as well as “”chaperones” and “chaperons”. I suggest to use “chaperone/chaperones”.
Ans: As per your suggestion we used “chaperone/chaperones” throughout the text.
- Both terms “ubiquitination” and “ubiquitinylation” are used in the manuscript. I suggest to use “ubiquitination” throughout the text as well as “ubiquitinated” instead of “ubiquitinylated”.
Ans: As per your suggestion we used “ubiquitination” and “ubiquitinated” throughout the text.
- Disease names should be given in the same way throughout the text and in the Abbreviations section (Parkinson's disease, Alzheimer's disease, Huntington’s disease or Parkinson disease, Alzheimer disease, Huntington disease).
Ans. Corrected.
- Page 2, lines 38-39: It should be “This quality control system includes the molecular chaperones, ubiquitin/proteasome-dependent protein degradation, and autophagy machinery” instead of “This quality control system includes the molecular chaperones, protein degradation, and autophagy machinery“.
Ans. Corrected.
- Page 2, lines 49-50: It should be “ubiquitin-dependent proteolysis” instead of “ubiquitin-dependent proteostasis”.
Ans. Corrected.
- Page 2, lines 68-69: The statement ”exposure to pH, temperature” is not clear enough.
Ans. Statement changed.
- Page 3, line 76: There is no information about TRiC in Ref. [10]. This reference is for the bacterial chaperonin GroEL. I suggest to include another reference for the eukaryotic chaperonin TRiC (for example the review: Balchin D, Hayer-Hartl M, Hartl FU. In vivo aspects of protein folding and quality control. Science. 2016 Jul 1;353(6294):aac4354. doi: 10.1126/science.aac4354. PMID: 27365453).
Ans. Reference added.
- Page 3, lines 83-89: The fragment about Hsp70 and J-proteins should be checked and corrected (see also comments no. 11 and 12). I suggest to include a more recent reference on Hsp70 (for example: Mayer, M.P.; Gierasch, L.M. Recent advances in the structural and mechanistic aspects of Hsp70 molecular chaperones. J. Biol. Chem. 2019, 294, 2085–2097).
Ans. Corrected and Added.
- Page 3, line 84: I suggest “a co-chaperone known as J-protein (also called Hsp40)” instead of “a co-chaperone known as J-protein”.
Ans. Corrected.
- Page 3, lines 88-89: The statement “Hsp70 co-chaperone and Hsp40 have a domain known as the J-protein binding domain ” is not clear. Hsp40 co-chaperones are collectively called J-proteins because they all contain a J-domain that is required for interaction with Hsp70. I suggest to check Ref. [13] as well as Frontiers in Neuroscience 2018, vol.11, article 743 for information about Hsp70 and Hsp40 interaction. Ref. [15] seems not adequate.
Ans. Corrected.
- Page 4, lines 113 and 115: Ref. [12] seems not adequate.
Ans. Corrected.
- Page 4, lines 124-126: Not all co-chaperones mentioned are included in Table 1.
Ans. Table 1 included only those Co-chaperones which have role in the protein degradation.
- Page 5, lines 149-151: The statement “E3 ligases further degrade the misfolded proteins by tagging them for ubiquitinylation and direct them towards proteasome machinery” is not entirely correct because E3 enzymes (ubiquitin ligases) do not degrade proteins. Proteins are degraded by 26S proteasomes.
Ans. Corrected.
- Page 7, lines 240-242: The sentence beginning with “p97 (VCP)” is difficult to understand.
Ans. Sentence is changed.
- Page 7, lines 244-246: I suggest to check and rewrite the sentence beginning with “26S proteasome (26S proteasome is mentioned as the proteasomal enzyme together with subunits of 19S regulatory complex and cytosolic PNGase).
Ans. Corrected.
- Page 8, lines 260-262: The statement “One hypothesis targets its downstream toxicity on UPS. UPS is the site for protein degradation.[55-57] Several studies highlight its toxicity in proteasomal dysfunction.” is not clear enough. One can conclude from this fragment that “Several studies highlight UPS toxicity in proteasomal dysfunction.
Ans. Corrected.
- Page 8, line 273: The statement “affect 26S proteasome subunit” is not clear enough. The 26S proteasome is a multisubunit complex.
Ans. Corrected.
- Page 8, line 277: The statement “E1 ligase” seems not correct although it is taken from Ref. [62]. I would recommend to use “E1 enzyme” or “ubiquitin-activating enzyme (E1)” (see also comment no.2).
Ans. Term “E1 ligase” is replaced with “ubiquitin-activating enzyme (E1)”.
- Page 8, lines 288-290: The statement “Another metal gallium inhibits cellular 26S proteasome and chymotrypsin-like activity of the purified 20S proteasome with IC50values of 46, 27, and 16 μmol/L.[65]” is not clear enough. The IC50 values mentioned are those reported for three of five gallium complexes tested by [65].
Ans. Corrected.
- Page 11, lines 377-380: It is difficult to understand the statements “Current research has revealed that autophagy can be an alternative or accompanying process to apoptosis in As-exposed cells.[86] examined the capability of As2O3 in inducing autophagic cell death in leukemic cell lines: Their data demonstrate that As is a potent inducer of autophagy.[87]”. This fragment should be corrected (see also comment no. 1).
Ans. Corrected.
- Page 11, Table 2 (third column/Chlorpyrifos row): The statement “Increase in Autophagy by affecting autophagy” is not clear.
Ans. Corrected.
- Correcting and editing the English used in the manuscript (including the Graphical Abstract) is recommended (some suggestions are given in comments).
Ans. Corrected.
Minor comments:
- Page 2, line 52: It should be “cataract, and diabetes” instead of “Cataract, and Diabetes”.
Ans. Corrected.
- Page 2, line 67: It should be “Aberrant protein processing by molecular chaperones” instead of “Aberrant protein processing by molecular chaperon”.
Ans. Corrected.
- Page 3, line 101: It should be “native” instead of “naive”.
Ans. Corrected.
- Page 4, lines 117 and 119: It should be “BAG1” instead of “Bag1” (this name should be given in the same way throughout the text).
Ans. Corrected.
- Page 4, line 122: It should be “Hsp70 and Hsp90” instead of “HSP70 and HSP90”.
Ans. Corrected.
- Page 4, line 124: It should be “HSJ1a” instead of “HSJ1A”, “BAG6” instead of “BAG 6”, and “UFD2B” instead of “UFD2D”.
Ans. Corrected.
- Page 4, TABLE 1: First column (SN) is not necessary. It should be “Hsp70” instead of “HSP70”, “Hsp90” instead “HSP90”, “RING” domain” instead of “Ring domain”, “BAG domain” instead of “Bag domain”, and “HSJ1a” instead of “HSJ1A”.
Ans. Corrected.
- Page 5: Figure 1 should be placed within the section 2.3.
Ans. It is included in section 2.3.
- Page 6, line 170: It should be “The IPOD sequestered proteins are” instead of “The IPOD sequestered proteins are”.
Ans. Corrected.
- Page 7, lines 227-228: The term “ERAD” should be included in the text when it arrives for the first time (see line 220).
Ans. Corrected.
- Page 7, line243: I suggest “ubiquitinated substrate” instead of “ubiquitinylated substrate” (see comment no. 4).
Ans. Corrected.
- Page 7, line 246: It should be “associated” instead of “associate”.
Ans. Corrected.
- Page 8, line 256: I suggest a new paragraph from the sentence beginning with “Rotenone, an organic pesticide”.
Ans. Corrected.
- Page 10, line 334: It should be ‘pyrethroid-associated toxicity” instead of “pyrethroid associate toxicity”.
Ans. Corrected.
- Page 10, line 349: It should be “unfolded protein response in ER (UPR)” instead of “unfolded protein response in ER” (The Authors use the UPR abbreviation in the next sentence).
Ans. Corrected
- Page 10, line 354: It should be “activation of UPR [80].” instead of “activation of unfolded protein response (UPR).[80]”.
Ans. Corrected.
- Page 10, line 355: I suggest a new paragraph from the sentence beginning with “During ERAD process”.
Ans. Corrected.
- Page 10, line 362: It should be “UBI4” instead of “UBI14” (There is an error in the abstract of Ref. [81] taken by the Authors as a source of information).
Ans. Corrected.
- Page 10, line 364: It should be “on autophagy” instead of “on Autophagy”.
Ans. Corrected.
- Page 11, line 379: It should be “As2O3” instead of “As2O3”.
Ans. Corrected.
- Page 11, Table 2 (first column): It should be “As2O3” instead of “AS2O3”.
Ans. Corrected.
- Page 11, Table 2 (third column): I suggest “the fusion oncoprotein PML/RARA” instead of “the fusion protein PML/RARA”.
Ans. Corrected.
- Page 11, Table 2 (third column/Chlorpyrifos row): It should be “Increase in autophagy” instead of “Increase in Autophagy”.
Ans. Corrected.
- Page 11, Table 2 (second column/Nickel row): I suggest “Human bronchial epithelial BEAS-2B cells” instead of “BEAS-2B”.
Ans. Corrected.
- Page 11, Table 2 (second column): I suggest “human neuroblastoma SH-SY5Y cells” instead of “SH-SY5Y”.
Ans. Corrected.
- Page 11, Table 2 (second column/Glyphosate row): I suggest “neuronal differentiated PC12 cells” instead of “PC12”.
Ans. Corrected.
- Page 12, line 407: I suggest “ABBREVIATIONS” instead of “ABBREVIATION”.
Ans. Corrected.
- Page 12, line 408: It should be “HSP, heat shock protein” or “HSPs, heat shock proteins” instead of “HSP, heat shock proteins”.
Ans. Corrected.
- Page 12, lines 408-415: I suggest an alphabetical order of abbreviations. “UPR” and “CSE” should be included.
Ans. Corrected
- Page 12, line 414: It should be “DE, diesel exhaust” instead of “DE, Diesel exhaust”.
Ans. Corrected.
Reviewer 2 Report
The review article describes the cellular machinery that corrects and/or rejects misfolded/oxidized proteins, whether the functions of this machinery are adequate/effective, and how they are compromised by various classes of pollutants in their association with various environmental diseases in man. Other than needed minor editing throughout the article and possibly in its title (shown in the attached article file), the manuscript is concisely written and the conclusions drawn are supported by the data and the adequate referencing of past studies. The study is scientifically sound and advances further the knowledge in this very important area of modern biochemistry.

Author Response
First, we would like to express our sincere gratitude to the reviewers for their critical and insightful suggestions and comments. We revised the manuscript according to these suggestions. All amendments are written in blue in the revised manuscript.
The review article describes the cellular machinery that corrects and/or rejects misfolded/oxidized proteins, whether the functions of this machinery are adequate/effective, and how they are compromised by various classes of pollutants in their association with various environmental diseases in man. Other than needed minor editing throughout the article and possibly in its title (shown in the attached article file), the manuscript is concisely written and the conclusions drawn are supported by the data and the adequate referencing of past studies. The study is scientifically sound and advances further the knowledge in this very important area of modern biochemistry.
Ans. Thanks for your appreciation, we have done the required editing as per your suggestion.
Round 2
Reviewer 1 Report
The revised manuscript has been significantly improved according to reviewers’ suggestions and comments.